# Effect of the Zirconia Particle Size on the Compressive Strength of Reticulated Porous Zirconia-Toughened Alumina

Chae-Young Lee [1,2], Sujin Lee [1], Jang-Hoon Ha [1,*,†], Jongman Lee [1], In-Hyuck Song [1] and Kyoung-Seok Moon [2,*,†]

1 Ceramics Materials Division, Korea Institute of Materials Science, 797 Changwondaero, Seongsan-gu, Changwon 51508, Korea; cy916@kims.re.kr (C.-Y.L.); adsj1503@kims.re.kr (S.L.); jmlee@kims.re.kr (J.L.); sih1654@kims.re.kr (I.-H.S.)
2 School of Materials Science and Engineering, Gyeongsang National University, Jinju 52828, Korea
* Correspondence: hjhoon@kims.re.kr (J.-H.H.); ksky.moon@gnu.ac.kr (K.-S.M.)
† These authors contributed equally to this work.

**Featured Application: Ceramic Membranes.**

**Abstract:** Reticulated porous ceramics have attracted researchers owing to the separation and collection properties of porous materials and the combined high thermal resistance and chemical stability of ceramics. Among various kinds of reticulated porous ceramics, we investigated the feasibility of using reticulated porous Zirconia-toughened Alumina as applications such as dielectric barriers, insulators, and filters with acceptable properties. An acceptable range of the compressive strength for reticulated porous ZTA applications is approximately 1 MPa. However, when the pore density of the reticulated porous ZTA specimen prepared using coarse zirconia was 60, maximum compressive strength of 1.63 MPa was obtained. To enhance the compressive strength of reticulated porous ZTA specimens, rheological control of the ZTA slurry is most important by optimizing the viscosity of the ZTA slurry, and the composition (average particle size, solid loading, organic binder, and thickener) of the ZTA slurry was controlled. The optimized processing conditions to enhance the compressive strength of reticulated porous ZTA specimens were determined. Consequently, we enhanced the compressive strength of the reticulated porous ZTA specimens from 0.37 MPa to 3.11 MPa by optimizing the ZTA slurry when the solid loading content, the pore density, the sintering temperature, the amount of PVA, and the amount of thickener were 66 wt.%, 60 PPI, 1600 °C, 2 wt.%, and 0.15 wt.%, respectively.

**Keywords:** reticulated porous ZTA; optimized viscosity; rheological control; compressive strength





## 1. Introduction

Porous ceramics have certain unique properties of porous materials, such as collection/separation of open pores and heat-shielding/sound-absorbing of closed pores. These properties cannot be achieved by conventional dense materials. In addition, the characteristics of porous ceramics are such that they cannot be realized with porous polymers or porous metals. Among the various types of porous ceramics, reticulated porous ceramics have high porosity (≥90%) and permeability simultaneously [1]. In addition, these materials are characterized by a 3D network structure with applications to radar-absorbing materials [2], diesel particulate filters (DPF) [3], solid oxide fuel cell (SOFC) electrodes [4], catalyst carriers [5], and molten metal filtration filters [6].

There are various porous ceramic fabrication methods, including the replica template method [7,8], the direct foaming method [9,10], the partial sintering method [11,12], and the sacrificial template method [13,14]. Among these methods, the replica template method can be used to prepare reticulated porous ceramics. This method utilizes a sacrificial polymer template that is impregnated with a ceramic slurry, with the sacrificial polymer template then removed through a heat treatment process [15,16]. When preparing reticulated porous

ceramics using the replica template method, the process parameters include (1) the viscosity of the ceramic slurry and (2) the pore density (PPI) of the sacrificial polymer template. Reticulated porous ceramics having various properties can be prepared by changing the viscosity characteristics of the ceramic slurry and controlling the pore density of the sacrificial polymer template. Optimizing the ceramic slurry can enhance the compressive strength of reticulated porous ceramics. An optimized slurry is a slurry having adequate fluidity and viscosity. Specifically, it can be used to coat the strut walls deep inside of the sacrificial polymer template uniformly.

Zirconia-toughened alumina (ZTA) has a wide range of applications, and much experimental research has already been conducted. However, there are too few studies of reticulated porous ZTA. The mechanical properties of ZTA depend on the properties of the zirconia used, which are determined by the stabilizer content and type. In addition, ZTA properties are determined by the transformation ratio, alumina content, sintering temperature, and the interrelation between the density and the amount of tetragonal phase used [17,18]. When it contains 2–3 mol% of yttria as a stabilizer to have a tetragonal phase of 100%, it is easy to sinter and can be made to transition to the monoclinic phase by external stress, which causes micro-cracks to absorb thermal shocks and increase toughness [19]. In addition, yttria-stabilized zirconia (YSZ) containing 3–5 mol% of yttria has excellent strength compared to stabilized zirconia containing other oxides; the most stable yttria-stabilized zirconia is yttria-stabilized zirconia with 3 mol% of yttria added [20,21]. Therefore, in this study, 3 mol% YSZ was used, and reticulated porous ZTA was prepared using the replica template method. Generally, there is a limited range of average particle sizes that can be controlled with several methods such as ball milling. Therefore, we initially attempted to determine which size range of the zirconia raw material would be appropriate for use in this study. If the compressive strength of reticulated porous ZTA specimens prepared from nano-sized zirconia particles is appropriate, it would be necessary to purchase nano-sized zirconia with different average particle sizes and use it, as it is well known that the effect of ball-milling is marginally effective when the average particle size range approaches the nano-size range [22]. However, there are several problems when using various nano-sized zirconia particles from different producers. The average particle size of zirconia, depending on the manufacturing method and the shape of the particles, can also differ, increasing the number of factors to consider. Otherwise, it becomes necessary to fabricate reticulated porous ZTA specimens prepared from various types of micro-sized zirconia particles, which are controlled by ball-milling. Therefore, we purchased and used zirconia particles with the following two size ranges to create representative specimens. We used two different average particle sizes of 0.3–0.6 μm (called "coarse") and 40 nm (called "fine") to control the average particle size. In addition, it is known that ball-milling is effective down to sub-micron particles and does not play a significant role in nano-sized particles. Hence, coarse particles of a sub-micron size were ball-milled to control the average particle size. In addition, we attempted to enhance the compressive strength of reticulated porous ZTA by controlling the composition of the slurry and the pore density of the sacrificial polymer template. The changes of the viscosity and thereby the microstructure and compressive strength of the reticulated porous ZTA were investigated.

## 2. Materials and Methods

In this study, alumina with an average particle size of 0.26 μm (AKP-30, Sumitomo Chemical, Tokyo, Japan) was used as a matrix. Zirconia with 3 mol% of yttria with (1) 0.3–0.6 μm ("coarse", Qingdao Terio Corporation, Qingdao, China) and (2) 40 nm ("fine", US Research Nanomaterials, Houston, TX, USA) average particle sizes were added.

Commercial polyurethane foams (SKB Tech, Seoul, Korea) as a sacrificial polymer template with 25, 45, and 60 PPI (pores per inch) was used at a size of 20 mm × 20 mm × 20 mm. In a previous experiment [23], the zirconia content of 40 wt.% had the highest compressive strength. Therefore, the content of zirconia was fixed at 40 wt.%. In addition, the solid loading level (57–66 wt.%), thickener content (0.5–3.0 wt.%), and organic binder content

(0–10 wt.%) were adjusted to investigate the optimized composition of the ZTA slurry. The conditions under which the ZTA slurry was most optimized and the compressive strength of the reticulated porous ZTA specimen was highest are as follows.

### 2.1. Preparation of the ZTA Slurry by Coarse Zirconia Particles

The ZTA slurry used to coat the sacrificial polymer template contains alumina and zirconia at 67 wt.% (zirconia is 40 wt.% of alumina), 1 wt.% of Dolapix CE 64 as a dispersant (Zschimmer & Schwarz GmbH Co., Lahnstein, Germany), 2.5 wt.% of Methyl cellulose (MC) as a thickener (Sigma-Aldrich, Darmstadt, Germany), 100 mL of distilled water, and 10 wt.% of polyvinyl alcohol (PVA) as an organic binder (Mn 500, Junsei chemical, Tokyo, Japan).

### 2.2. Preparation of the ZTA Slurry by Fine Zirconia Particles

The ZTA slurry used to coat the sacrificial polymer template contains alumina and zirconia at 67 wt.% (zirconia is 40 wt.% of alumina), 1 wt.% of Dolapix CE 64 as a dispersant (Zschimmer & Schwarz GmbH Co., Germany), 0.15 wt.% of Methyl cellulose (MC) as a thickener (Sigma-Aldrich, Darmstadt, Germany), 100 mL of distilled water, and 2 wt.% of polyvinyl alcohol (PVA) as an organic binder (Mn 500, Junsei chemical, Japan).

### 2.3. Preparation of a Reticulated Porous ZTA Specimen

An impregnation process was applied to coat the prepared ZTA slurry onto the sacrificial polymer template, after which it was squeezed to remove any excess ZTA slurry remaining on the sacrificial polymer template. Subsequently, the binder was removed a by heat treatment at 400 °C for 2 h, and the specimens were sintered at 1600 °C for 3 h.

### 2.4. Characterization of a Reticulated Porous ZTA Specimen

The ZTA slurry viscosity levels were determined using a rotary viscometer (ViscoQC 300, Anton Paar GmbH, Graz, Austria). The particle size distribution of the ZTA was measured with a particle size analyzer (PSA) (LSTM 13 320 MW, Beckman Coulter, Pasedena, CA, USA). After preparing the reticulated porous ZTA specimens, the microstructures were characterized by scanning electron microscopy (JSM-6610LV, Jeol, Japan). The compressive strength capabilities of the reticulated porous ZTA specimens were measured with a tensile tester (RB302 Microload, R&B, Daejeon, Korea). To increase the accuracy of the experiment, all reticulated porous ZTA specimens were prepared 10 times and then analyzed.

## 3. Results

In this study, reticulated porous ZTA specimens with 25, 45, and 60 PPI (pores per inch) were prepared via the replica template method. The acronym for pores per inch, PPI, refers to the number of pores in the sacrificial polymer template. The higher the pores per inch (PPI) value is, the more pores there are and the smaller they are. Figure 1a shows optical images of sacrificial polymer template specimens at 25, 45, and 60 PPI. Figure 1b shows the reticulated porous ZTA specimens at 25, 45, and 60 PPI. These specimens were prepared via the replica method. Typical scanning electron microscopy images of the fractured strut wall of the reticulated porous ZTA specimen are shown in Figure 1c. A strut wall with a uniform thickness was formed.

We attempted to determine the size range of the zirconia raw material that would be appropriate for use in this study. Therefore, the zirconia particles used here have two size ranges as representative. We used two different average particle sizes of 0.3–0.6 μm (coarse zirconia) and 40 nm (fine zirconia) to control the average particle size. Scanning electron microscopy images of as-received coarse zirconia and fine zirconia particles are correspondingly shown in Figure 2a,b.

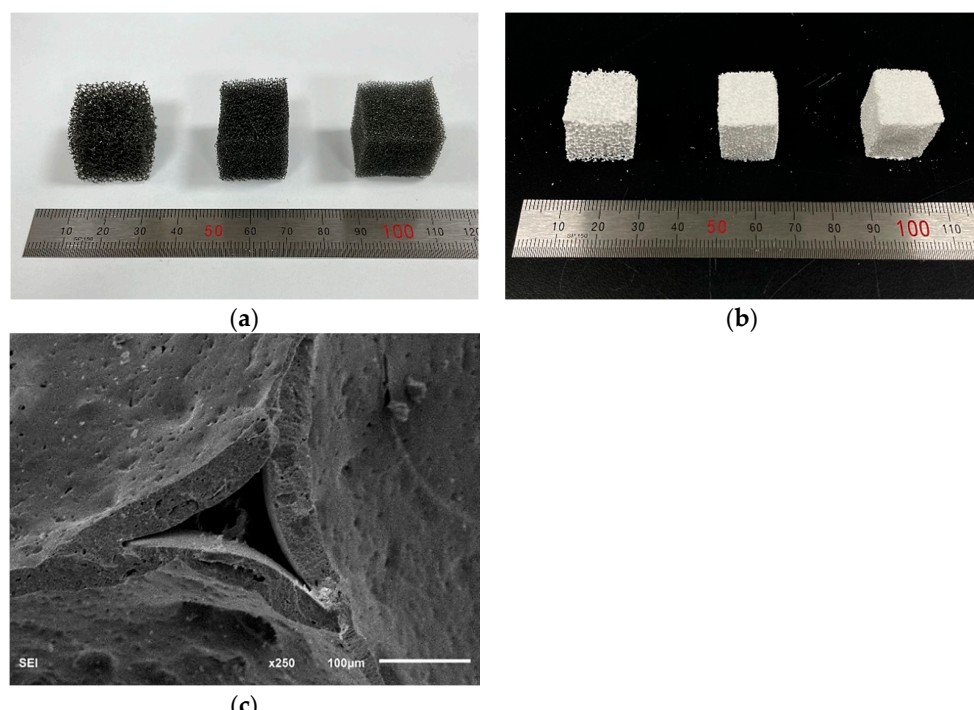

(a)

(b)

(c)

**Figure 1.** Optical images of (**a**) sacrificial polymer templates with pore densities of 25, 45 and 60, (**b**) reticulated porous ZTA specimens with corresponding pore densities of 25, 45 and 60, and (**c**) typical scanning electron microscopy image of the fractured strut wall of a reticulated porous ZTA specimen.

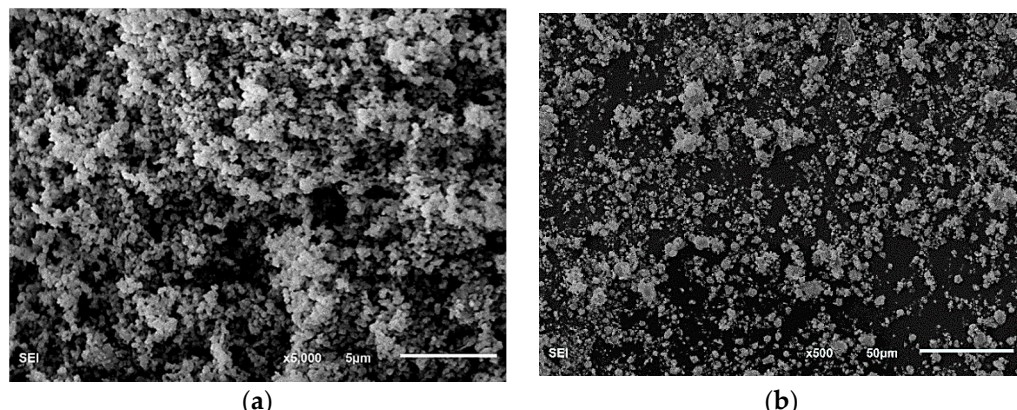

(a)

(b)

**Figure 2.** Typical scanning electron microscopy images of as-received zirconia particles: (**a**) coarse and (**b**) fine.

### 3.1. Reticulated Porous ZTA Specimen Prepared by Coarse Zirconia Particles

3.1.1. The Effect of the Solid Loading on the Reticulated Porous ZTA Specimen

Figure 3a,b show part of the preliminary experiment; the composition was selected by adjusting the solid loading level and the thickener content. The solid loading level was 66 wt.%, and this led to the highest compressive strength. Therefore, the amount of solid loading was fixed at 66 wt.%. However, the reticulated porous ZTA specimens had a very low compressive strength of approximately 0.22 MPa when the solid loading content was 66 wt.%. This limited the application of the reticulated porous ZTA, and when the solid loading content was 66 wt.%, the viscosity was not sufficient to coat the strut walls of the reticulated porous ZTA specimens tightly.

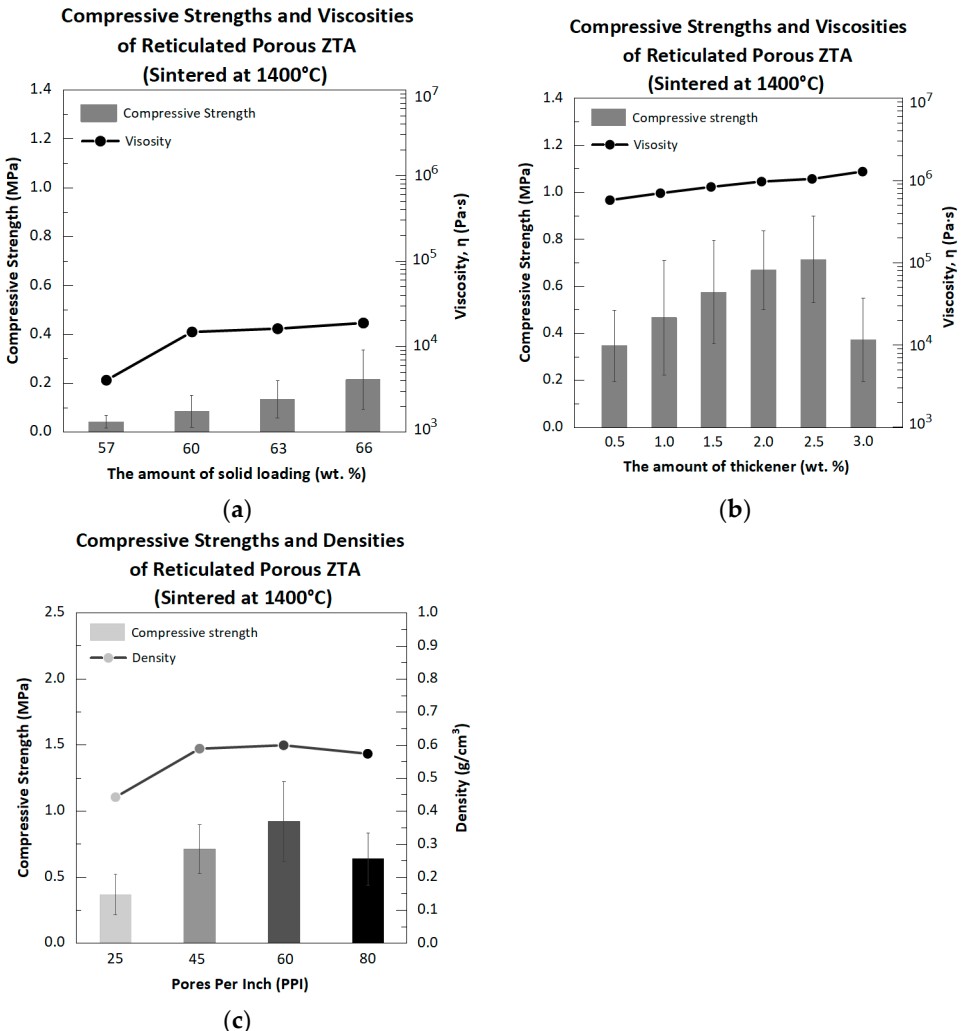

**Figure 3.** Viscosities at a shear rate of 0.01 s$^{-1}$ of the ZTA slurry, and the compressive strengths: (**a**) 57–66 wt.% solid loading, (**b**) 0.5–3.0 wt.% MC, and (**c**) compressive strengths and densities of reticulated porous ZTA specimens with pore densities of 25, 45, 60 and 80.

### 3.1.2. The Effect of the Thickener on the Reticulated Porous ZTA Specimen

Controlling the viscosity by managing the solid loading level only requires additions of alumina and zirconia particles to the point that the desired viscosity is reached, but this approach is not efficient. Therefore, the solid loading content was fixed at 66 wt.%, and the viscosity was controlled by adding a thickener. The thickener acts to increase the uniformity when mixed with water [24]. By suppressing the sedimentation of the slurry, the viscosity of the upper and lower positions of the slurry and the dispersion state remain the same for a long period of time to stabilize the viscosity of the slurry. In addition, the thickener improves the mechanical strength of the material by increasing the thickening properties, preventing separation between materials, and retaining the water necessary for hydration [23]. In addition, when polyvinyl alcohol as an organic binder and methyl cellulose as a thickener are used together, more effective improvement of the compressive strength becomes possible [24,25]. Therefore, the solid loading content was fixed at 66 wt.%, and the viscosity was controlled with the addition of a thickener, as mentioned above. When the thickener was added, the viscosity increased by approximately 100 times, as shown in Figure 3b. The thickener content was fixed at 2.5 wt.% and led to the highest compressive strength given the optimized viscosity.

The compressive strength outcomes at pore densities 25, 45, 60, and 80 for the reticulated porous ZTA specimens are shown in Figure 3c. As the PPI was increased, the pore

density of the reticulated porous ZTA specimens increased, resulting in higher compressive strength. However, when the pore density was 80, the viscosity of the ZTA slurry was high, and the compressive strength decreased. As the pore density of the reticulated porous ZTA specimens approaches 80, preparation becomes difficult owing to the high pore density of the sacrificial polymer template, also making it difficult to remove the residual slurry deep inside the sacrificial template. In addition, closed pores form inside the sacrificial polymer template due to residual slurry after the squeezing process, thereby reducing the compressive strength [26]. Therefore, it was confirmed that the reticulated porous ZTA specimens can be prepared up to a pore density of 60.

### 3.1.3. The Effect of the Sintering Temperature on the Reticulated Porous ZTA Specimen

Figure 4a shows the compressive strength outcomes of reticulated porous ZTA specimens at 25, 45, and 60 PPI according to the sintering temperature. It can be seen that the compressive strength increased as the sintering temperature and PPI were increased. The reticulated porous ZTA specimens at a pore density of 60 sintered at 1600 °C had the highest compressive strength (approximately 1.63 MPa). The outcomes with pore densities of 25, 45, and 60 of the reticulated porous ZTA specimens according to the sintering temperature are shown in Figure 4b. Overall, the trend for the densities was similar to that of the compressive strength. As the sintering temperature and pore density of the reticulated porous ZTA specimens were increased, the density increased. Figure 4c shows XRD patterns of the phases of reticulated porous ZTA with different sintering temperatures. For zirconia sintered at 1400, 1500, and 1600 °C, the main phase was the tetragonal phase, while the monoclinic phase of zirconia was not observed. In addition, as the sintering was temperature increased, the peak intensity of the tetragonal phase increased. Figure 4d shows the linear shrinkage outcomes as the sintering temperature was increased. As the sintering temperature was increased, the shrinkage rate of the reticulated porous ZTA increased. This is considered to stem from the number of open pores, which decreased due to the greater densification between the alumina and zirconia particles as the sintering temperature was increased. This can also be explained by the microstructure (Figure 5a–c).

Typical backscattered electron images of the reticulated porous ZTA specimens when sintered at 1400, 1500, and 1600 °C, prepared from coarse zirconia particles, are shown in Figure 5a–c, respectively. The lighter color denotes the zirconia particles. It can be confirmed that the average particle size of alumina and zirconia increased as the sintering temperature was increased. When the reticulated porous ZTA was sintered at 1400 °C, there were many open pores between the alumina and zirconia particles, and densification was not achieved owing to the presence of many pores. Open pores between the particles and densification affect the compressive strength. The process of sintering can be explained by dividing it into the initial stage, intermediate stage, and final stage. In the initial stage of sintering, the interface of the particles to each other adheres to form a neck. In the intermediate stage of sintering, the pores gradually become round and grain growth occurs, resulting in larger particles than the raw material powder. After that, at the final stage of sintering, the pores become spherical and close, resulting in closed pores and additional grain growth. In addition, the higher the sintering temperature, the faster the sintering reaction occurs [27]. The microstructures when the reticulated porous ZTA is sintered at 1400, 1500, and 1600 °C correspond to the initial stage, intermediate stage, and final stage of the sintering process, respectively. Figure 4a shows that the compressive strength of the reticulated porous ZTA specimens was lowest when sintering was conducted at 1400 °C. When the reticulated porous ZTA specimens were sintered at 1500 °C, the number of open pores decreased and the grain boundaries between the alumina and zirconia particles became connected. When the reticulated porous ZTA specimens were sintered at 1600 °C, the grain size grew because there were no pores interfering with the grain growth process. These specimens showed the highest compressive strength due to the highest densification between the alumina and zirconia particles. Therefore, we set the sintering temperature to 1600 °C.

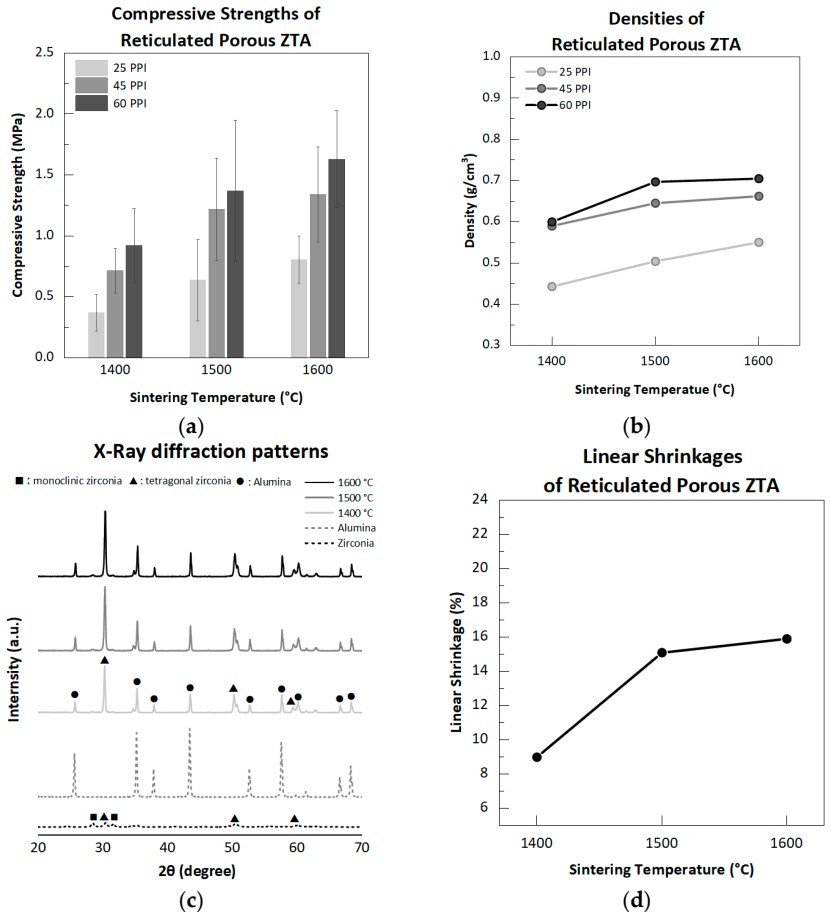

**Figure 4.** (**a**) Compressive strength outcomes of the reticulated porous ZTA specimens prepared from coarse zirconia particles sintered at 1400–1600 °C, (**b**) densities, (**c**) X-ray diffraction patterns of reticulated porous ZTA with different sintering temperatures, and (**d**) linear shrinkage rates.

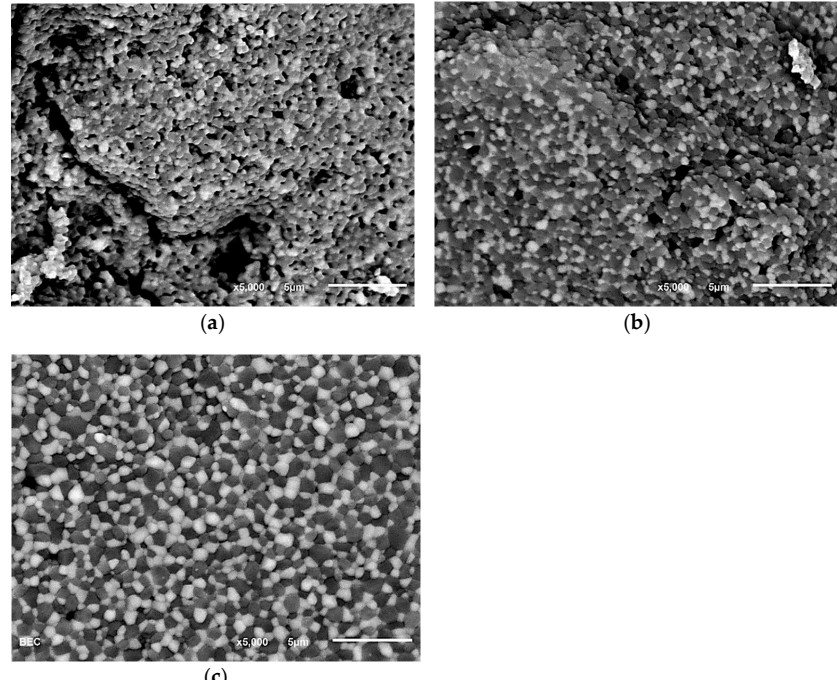

**Figure 5.** Typical back scattered electron images of reticulated porous ZTA specimens prepared using coarse zirconia when sintered at (**a**) 1400 °C (**b**) 1500 °C (**c**) 1600 °C.

### 3.1.4. The Effect of the Average Particle Size of ZTA on the Reticulated Porous ZTA Specimen

Generally, there is a limited range of average particle sizes that can be controlled with several methods, such as ball-milling. It is well known that the effect of ball-milling is marginally effective when the average particle size range approaches the nano-scale [22]. Therefore, the average particle size was controlled by ball-milling coarse zirconia particles in the micro-size range. Figure 6a shows the particle size distributions with various average particle sizes, which were controlled by adjusting the ball-milling time. The average particle size is a process parameter used to obtain the optimal composition. As the ball-milling time was increased, the average particle size decreased to 0.267 µm, 0.181 µm, 0.152 µm, and finally to 0.100 µm.

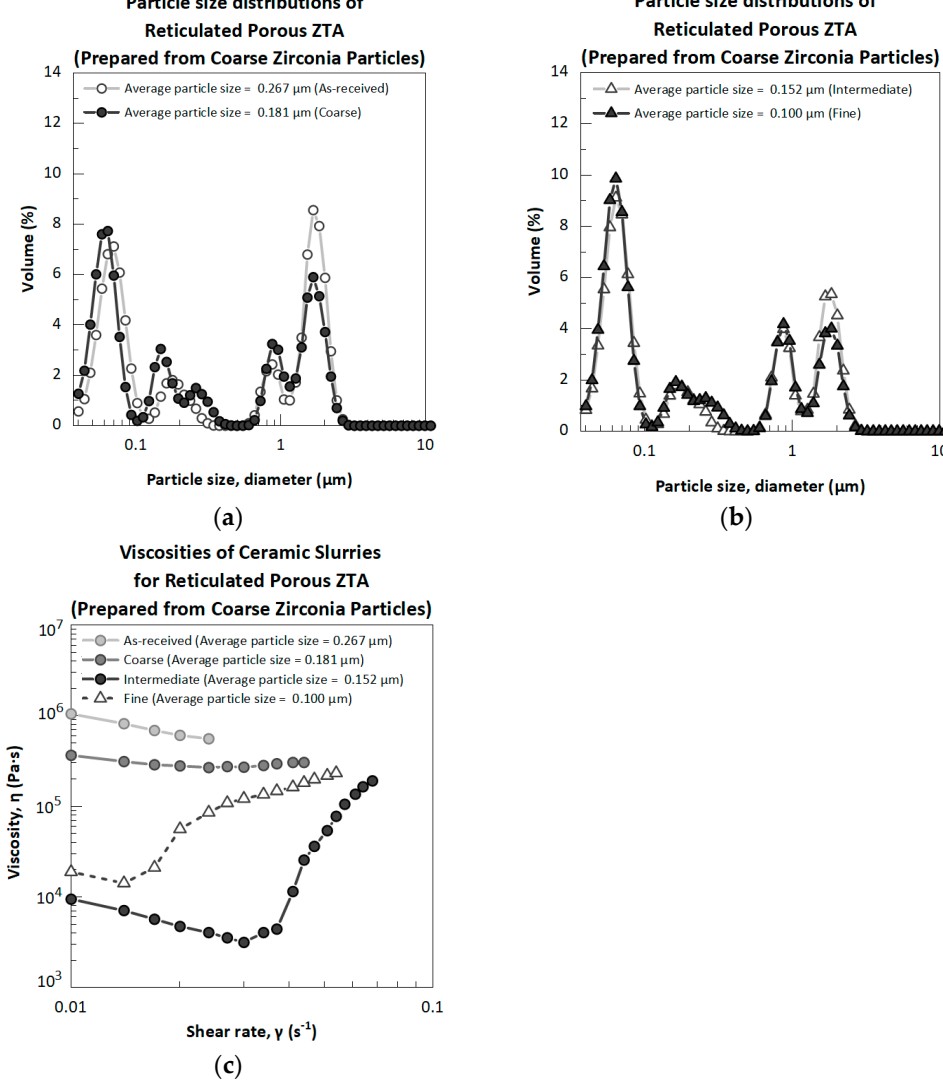

**Figure 6.** Particle size distributions of ZTA prepared from coarse zirconia: (**a**) as-received and coarse, (**b**) intermediate and fine cases, and (**c**) viscosities of ZTA slurries used to prepare the reticulated porous ZTA: as-received, coarse, intermediate, and fine cases.

The proper viscosity of the ZTA slurry is most important when preparing a reticulated porous ZTA specimen with a uniform structure. However, there are several problems that can arise during the preparation of reticulated porous ZTA specimens under certain conditions. If the viscosity of the ZTA slurry is too high, the ZTA slurry cannot penetrate deep inside the reticulated porous ZTA specimen. Therefore, voids are generated inside the reticulated porous ZTA specimen, which reduces the compressive strength. In contrast,

if the viscosity of the ZTA slurry is too low, the coating layer of the strut walls of the reticulated porous ZTA specimen becomes very thin and non-uniform. This causes the number of defects and cracks to increase along the strut walls of the reticulated porous ZTA specimens, which reduces the compressive strength. Figure 6b shows the viscosities of the ZTA slurries used here to prepare the reticulated porous ZTA specimens. When the average particle size is reduced due to ball-milling, the specific surface area increases, which causes more particles to adhere to the PVA, thus increasing the viscosity. Overall, however, as the average particle size is reduced, the slurry viscosity tends to increase. This is considered to stem from the excellent dispersing ability of the Dolapix CE 64 dispersant. The Dolapix CE 64 dispersant has a carboxyl group in its molecular structure. As the pH increases, dissociation of the carboxyl group is accelerated, and the molecules of the dispersant become negatively charged and adsorbed onto the oxide surface to form repulsive force between the particles [28]. The Dolapix CE 64 dispersant has a low molecular weight (~660 g/mol) compared to other types of dispersants and a pH of approximately 9, leading to better absorption of the particles in concentrated colloidal systems. It also completely dissociates in a solvent and has an electrostatic stabilizing effect immediately after it is added [28].

Figure 7a shows the compressive strength outcomes of reticulated porous ZTA specimens with various average particle sizes, all of which were controlled by the ball-milling time. As the average particle size of the raw material was decreased, the compressive strength of the reticulated porous ZTA specimens was enhanced compared to the as-received condition. However, it was found that the compressive strength of reticulated porous ZTA specimens decreases when the average particle size is determined by the use of fine particles. This is judged by re-aggregation due to over-milling. The agglutinability increases as the ball-milling continues because the particles become monodispersed, resulting in a decrease in the particle size, an increase in specific surface area, and unstable particles due to over-milling [29]. This can also be explained by the microstructure (Figure 7c). Figure 7c shows a typical back scattered electron image of a reticulated porous ZTA specimen when the average particle size was fine. The lighter color represents the zirconia particles. It can be seen that alumina and zirconia particles are agglomerated. A smaller average particle size increases the surface energy, causing densification, but also causes aggregation due to electrostatic attraction. Consequently, it can be seen that the reticulated porous ZTA specimens having a wide range of compressive strength values of 0.80~2.64 MPa can be prepared by controlling the average particle sizes. The density values of reticulated porous ZTA specimens are shown in Figure 7b. Overall, the trend here is similar to that of the compressive strength.

When preparing reticulated porous ZTA specimens using the replica template method, the important process parameters include the viscosity of the ZTA slurry and the pore density of the sacrificial polymer template. However, there is a limited value of the compressive strength that can be obtained by controlling the composition (solid loading, thickener, dispersant) of the ZTA slurry and the pore density of the sacrificial polymer template. In previous experiments [30,31], the content of the dispersant was 1 wt.%, and the compressive strength value was best under this condition. Therefore, in this study, the content of the dispersant was fixed at 1 wt.%, and it was not effective in terms of the time and cost to proceed with the experiment by controlling the content of the dispersant. The maximum compressive strength that can be obtained by adjusting the solid loading level and the thickener of the ZTA slurry, as well as the pore density of the sacrificial polymer template, was 2.64 MPa, with the samples prepared with coarse zirconia particles.

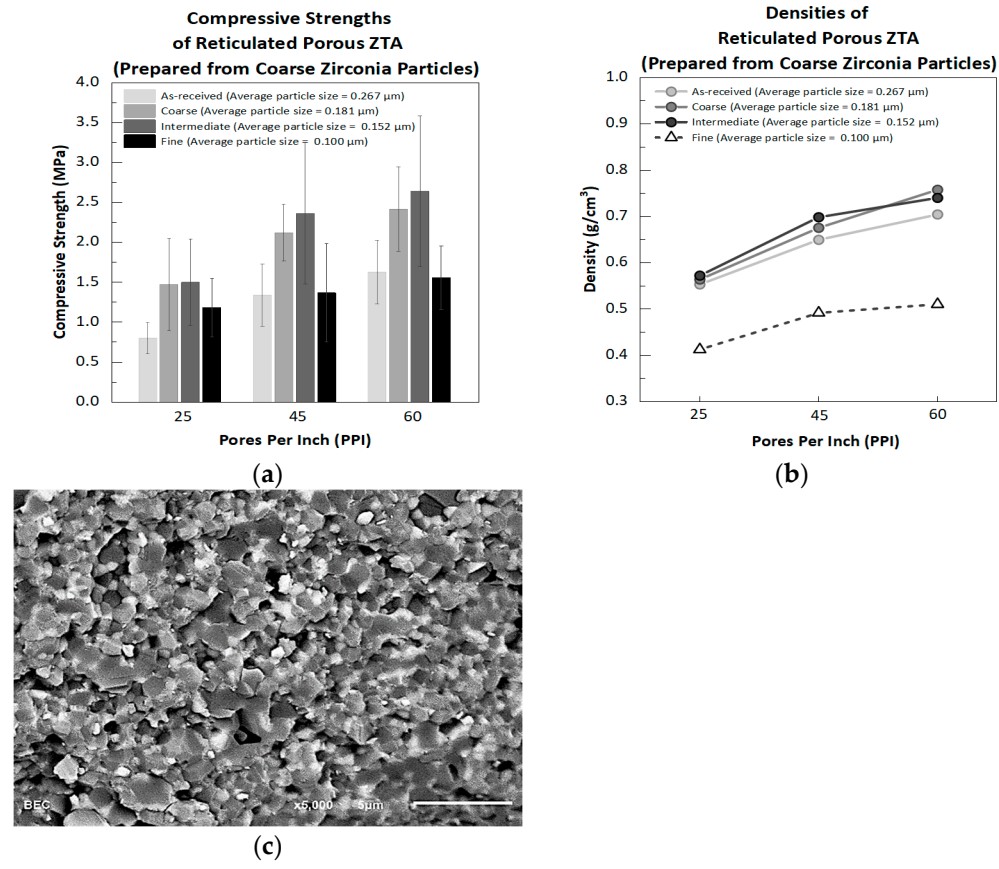

**Figure 7.** (**a**) Compressive strengths of reticulated porous ZTA specimens with various average particle sizes as controlled by adjusting the ball-milling time and (**b**) density, and (**c**) typical back scattered electron image of reticulated porous ZTA specimens when the average particle size was fine.

### 3.2. Reticulated Porous ZTA Specimen Prepared by Fine Zirconia Particles

To enhance the compressive strength of the reticulated porous ZTA specimens further, fine (nano-sized) zirconia particles were introduced. It is known that nano-sized particles significantly enhance the conventional mechanical properties owing to their relatively large specific surface area per unit volume at the same volume [32]. However, as the particle size reaches the nanoscale, the distance between the particle surfaces becomes shorter, causing aggregation. In addition, an effective mechanical dispersion method (such as ultrasound or agitation) with sub-micron particles does not play a major role when used with nano-sized particles. Therefore, in this study, a method capable of adsorbing a polymer ultimately to provide a barrier against aggregation [33] was used as a representative method of preparing a uniformly dispersed slurry. If the adsorption properties are well controlled, effective dispersion can be achieved.

#### 3.2.1. The Effect of the Organic Binder on the Reticulated Porous ZTA Specimen

In this study, polyvinyl alcohol (PVA) as an organic binder was added to prepare a uniform zirconia slurry. PVA increases the viscosity of the slurry and prevents the formation of cracks after the coating step [33]. PVA molecules enclose the particles and separate them from other particles, preventing agglomeration between the particles. In addition, PVA particles bond to each other as a matrix to maintain the shape of the sacrificial polymer template [34,35].

The viscosity levels of the ZTA slurries with the amount of PVA are shown in Figure 8a. As the amount of PVA was increased, the viscosity of the ZTA slurries increased. Figure 8b shows the compressive strengths and densities according to the amount of PVA. PVA at 2 wt.% led to the highest compressive strength (approximately 2.43 MPa). PVA assists

with contact between the alumina and zirconia particles, making them more viscous and preventing cracks or delamination that may occur during the deposition of the coating layer. However, excessive amounts of PVA cause agglomeration of alumina and zirconia particles. This can lead to an increase in the surface roughness of the coating layer. Therefore, it is important to add an appropriate amount of PVA. Reticulated porous ZTA with a uniform structure can be prepared when using an appropriate amount of PVA. This phenomenon is also seen in the microstructures.

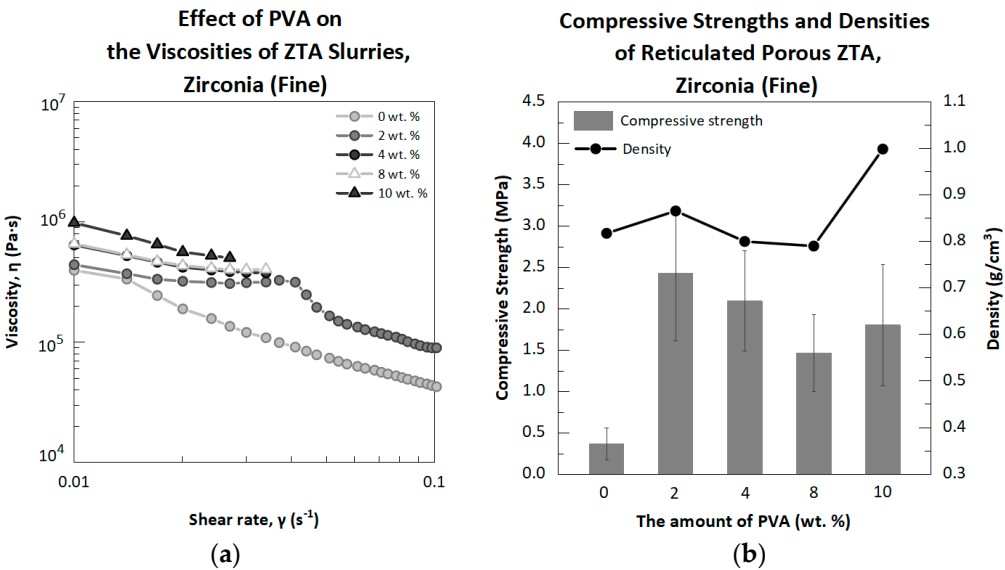

**Figure 8.** (**a**) Viscosities of the ZTA slurries with the amount of PVA (**b**) compressive strengths and densities with the amount of PVA.

Figure 9a shows a typical scanning electron microscopy image of the strut walls of a reticulated porous ZTA specimen without PVA added. It can be seen that the coating layer of the strut walls is very thin, and there are many cracks and defects. These cracks and defects affect the compressive strength. The reticulated porous ZTA specimens prepared without the addition of PVA had a very low compressive strength value of 0.37 MPa. Figure 9b,c show typical backscattered electron images of reticulated porous ZTA specimens when the amount of PVA was 2 wt.% and 10 wt.%, respectively. When the amount of the PVA was 2 wt.%, it was confirmed that fine zirconia particles are uniformly dispersed. Fine zirconia particles are uniformly dispersed, and the grain boundaries between the particles are clearly separated because a uniform PVA adsorption layer was formed on the surfaces of the zirconia and alumina particles to block aggregation effectively between the particles and prevent grain growth. In addition, the uniformity was enhanced, indicating a denser particle arrangement. The enhanced uniformity is initially due to the size of the particles to be filled, which is effectively reduced due to the blocking of agglomeration, thus improving the dispersibility of the slurry. Another cause is the PVA, which is uniformly adsorbed on the particle surface and which increases the fluidity of the slurry such that rearrangements between particles can occur relatively well [36].

However, when the amount of the PVA is 10 wt.%, it can be seen that fine zirconia particles were agglomerated. This affects the compressive strength (approximately 1.81 MPa). The amount of PVA at 10 wt.% had the highest density of approximately 1.0 g/cm$^3$ because the ZTA viscosity was too high, making it difficult to prepare the reticulated porous ZTA specimens. ZTA slurry when the viscosity was too high becomes thickly attached to the surface of the sacrificial polymer template. In addition, this form of ZTA slurry becomes trapped inside the sacrificial polymer template and cannot be removed. Therefore, when the compressive strength was higher than 8 wt.% PVA, the highest density resulted due to the blocking of the internal pores of the sacrificial polymer template. If the viscosity of the

ZTA slurry is too high, blocked pores arise inside the sacrificial polymer template, which greatly reduces the two main advantages of reticulated porous ceramics: low density and high permeability. Therefore, in this study, the amount of PVA at 2 wt.% was appropriate.

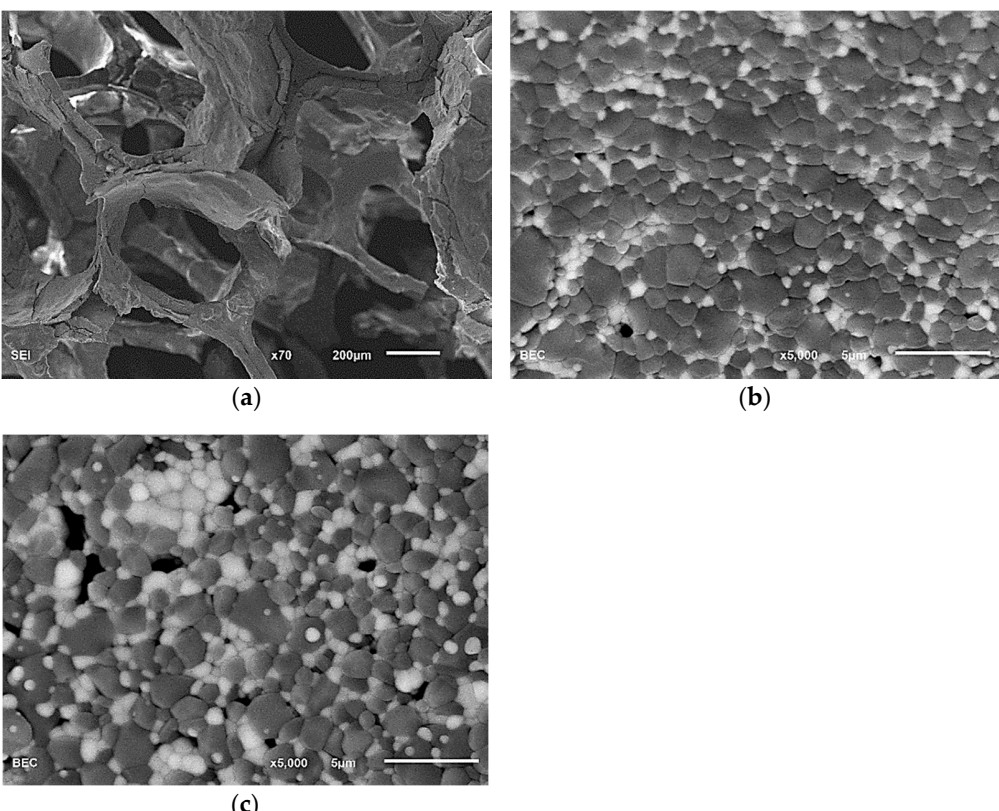

**Figure 9.** (**a**) Typical scanning electron microscopy image of the strut walls of a reticulated porous ZTA specimen without PVA, and (**b**) typical backscattered electron images of reticulated porous ZTA specimens when the amount of PVA at 2 wt.% and (**c**) at 10 wt.%.

### 3.2.2. The Effect of the Thickener on the Reticulated Porous ZTA Specimen

The viscosity levels as a function of the shear rate of ZTA slurries with different amounts of thickener are shown in Figure 10a. When more thickener was added, the viscosity of the ZTA slurry increased more. Figure 10b shows the compressive strengths and densities according to the amount of thickener. It was confirmed that the compressive strength of the reticulated porous ZTA specimens increased as the amount of thickener was increased. However, when the thickener was added at 0.2 wt.%, the compressive strength of reticulated porous ZTA specimens decreased due to the high viscosity of the ZTA slurry. The amount of thickener at 0.15 wt.% led to the highest compressive strength of approximately 3.11 MPa. In addition, overall, the densities showed a tendency similar to that of the compressive strength. Consequently, we enhanced the compressive strength of the reticulated porous ZTA specimens from 0.37 MPa to 3.11 MPa by optimizing the ZTA slurry.

An acceptable range of the compressive strength for reticulated porous ceramic applications is approximately 1 MPa. Typical examples of the compressive strengths of reticulated porous ceramics are 1.30 MPa [30] and 2.72 MPa [31] (alumina), 0.85 MPa [37] and 1.26 MPa [38] (zirconia), and 1.45 MPa [39] and 2.15 MPa [40] (zirconia-toughened alumina). The maximum compressive strength (3.11 MPa) of the reticulated porous ZTA we obtained means that it can be used for a variety of reticulated porous ceramic applications such as dielectric barriers, insulators, and filters.

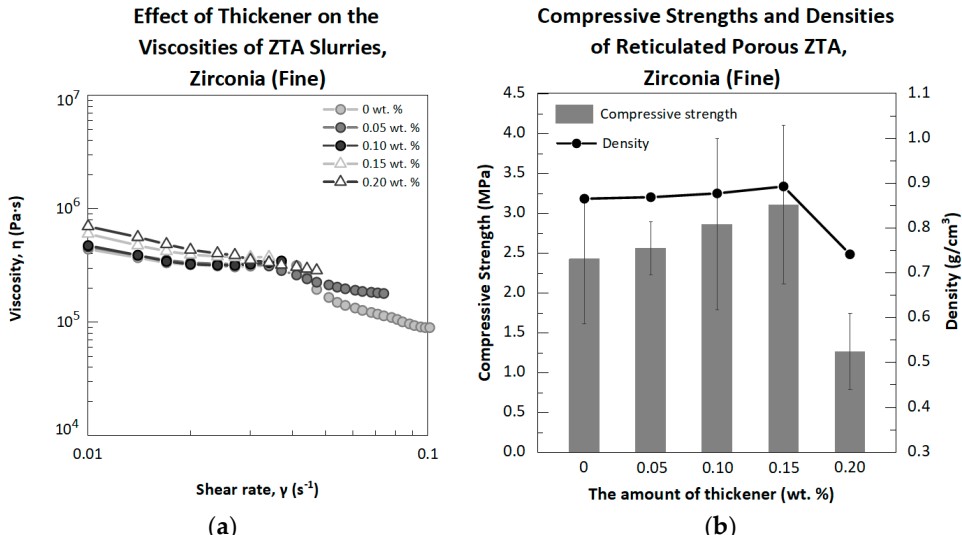

**Figure 10.** (**a**) Viscosities as a function of the shear rate of ZTA slurries with different amounts of thickener, and (**b**) compressive strength and density outcomes according to the amount of thickener used.

## 4. Conclusions

In this study, we attempted to determine the appropriate size range of zirconia raw materials to adopt. We used two different average particle sizes of 0.3–0.6 μm (called "coarse") and 40 nm (called "fine") to control the average particle size. In addition, we discussed the optimal processing conditions to enhance the compressive strength of reticulated porous ZTA specimens for each zirconia particle size.

When the pore density of the reticulated porous ZTA specimen prepared using coarse zirconia was 60, maximum compressive strength of 1.63 MPa was obtained. To enhance the compressive strength by making the average particle size smaller, the particle size was controlled by ball-milling. As a result, the compressive strength of the reticulated porous ZTA specimens was improved to approximately 2.64 MPa. When the reticulated porous ZTA specimen was prepared using fine zirconia with the addition of PVA at 2 wt.%, the compressive strength outcome was approximately 2.43 MPa. To improve the compressive strength of the reticulated porous ZTA specimens further, MC as a thickener was introduced. When PVA as an organic binder and MC as a thickener were used together, it was found to be more effective for improving the compressive strength [24,25]. Interestingly, the compressive strength of the reticulated porous ZTA specimen was improved to approximately 3.11 MPa. Whether other thickeners, as well as MC, have the same effect on the compressive strength can be an interesting topic for further study. However, this is beyond the scope of the present study and requires a considerable amount of experimentation. Therefore, this is left as further study. Consequently, to enhance the compressive strength of reticulated porous ZTA specimens, it is effective to use a fine average particle size and a thickener. In addition, these outcomes of the reticulated porous ZTA mean that it can be used for a variety of reticulated porous ceramic applications such as dielectric barriers, insulators, and filters.

**Author Contributions:** J.-H.H. and K.-S.M. conceived of and designed the experiments, C.-Y.L. and S.L. performed the experiments; J.L. and J.-H.H. analyzed the data, I.-H.S. contributed in the areas of the reagents/materials/analysis tools, and C.-Y.L. wrote the paper. All authors have read and agreed to the published version of the manuscript.

**Funding:** This research received no external funding.

**Institutional Review Board Statement:** Not applicable.

**Informed Consent Statement:** Not applicable.

**Acknowledgments:** This research was funded by the Fundamental Research Program of the Korean Institute of Materials Science (KIMS) by grant number PNK8120 and by the National R&D Program through the National Research Foundation of Korea (NRF) funded by the Ministry of Science and ICT (2020M3H4A3106359).

**Conflicts of Interest:** The authors declare no conflict of interest.

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
