# Peer review of "Effect of the Zirconia Particle Size on the Compressive Strength of Reticulated Porous Zirconia-Toughened Alumina"

_applsci, doi:10.3390/app12052316_

Round 1

Reviewer 1 Report

In this paper, compressive strength of reticulated porous ZTA ceramics prepared by replica template method is reported. However, information on the porosity and pore size, which give serious influence to the compressive strength, is lacked. Also, many readers want to know the information. In addition, preparation conditions of individual slurries such as pH, zirconia particle size and additive amount of thickener and dispersant are unclear, and information on dispersion state of the alumina and zirconia particles in slurries, which give serious influence to packing and sintering of particles as well as the compressive strength, is a few. The dispersion state can be estimated roughly by the measuring the viscosity of the slurry. Therefore, I hope addition of the data and considerations.

Reviewer 2 Report

Title: Effect of the Zirconia Particle Size on the Compressive Strength of Reticulated Porous Zirconia-toughened Alumina

Manuscript ID: applsci-1590347

The article investigated the effect of some parameters on the compressive strength of Reticulated Porous Zirconia-toughened Alumina such as solid loading level, thickener content, sintering temperature. The results are interesting, however, the expression of the manuscript is unclear and needs to revise for better quality. Here are some comments:

  1. The abstract should be rewritten. The results/data of this study should be added in the abstract for the readers to have a clearer view of the research.
  2. Please define the acronym of SOFC on line 37.
  3. The Materials and Methods section must be rewritten. All experiments must be detailly described in this section such as (i) how many experiments were conducted; (ii) at what values the parameters (e.g., solid loading level, thickener content, etc.) were investigated; (iii) how many times the experiments were repeated; etc. Besides, an experiment diagram should be added to easily understand all experiments conducted in this study.
  4. The Results should be separated into some subsections corresponding to each experiment e.g., 3.1. The effect of solid loading on compressive strength of ZTA; 3.2. The effect of sintering temperature on compressive strength of ZTA, etc. The current presentation of the manuscript is difficult for the reader to get the information.
  5. Please cite the references of some explanation for the results mentioned in the manuscript. For example, cite references for the sentences on lines 156 – 161 (“As the pore density … the compressive strength”).
  6. Please uniform PPI value in the manuscript. PPI values were 25, 45, 60 on lines 114, 118, 119, 168, … but 25, 45, 60, 80 on line 152 and Figure 3c.
  7. The conclusion should be shortened and focus on the new findings of this study. The data should be presented in the abstract, not in the conclusion.
  8. Almost a third of the references are more than 15 years old. Please update the latest references.
  9. Please check the third reference. Is it a book/paper or a thesis/dissertation? If it is a thesis/dissertation, it will not be an official reference.

Reviewer 3 Report

The manuscript “Effect of the Zirconia Particle Size on the Compressive Strength of Reticulated Porous Zirconia-toughened Alumina” explores the suitable size range for the adoption of zirconia raw materials, the use of two different average particle sizes, 0.3-0.6 μm and 40 nm, to control the average particle size, and additionally the optimal processing conditions to increase the compressive strength of ZTA samples. In general, the manuscript is well written and suitable for publication in Applied Sciences. Some minor corrections could be taken into account and it needs clarification on certain issues as outlined below.

  • Have a relook on some grammatical errors available in the manuscript.
  • Improve Figures quality for the better eminence of your research work. especially in Figure 5, the size representations of the SEM images are not fully visible. Data out of the graph in Figures 6, 8a and 10a should be corrected.
  • In section 3,

It is clear that with the increase of the sintering temperature, neck formation occurs first (in Fig.5a) and the porous between these structures decreases with increasing temperature, forming larger particles (in Figs. 5b,5c). Brief information can be given to the reader about the sintering temperature and the growth mechanism of the grains.

  • The conclusion section is not suitable for literature comparison. The relevant section needs to be moved to section 3 and some content that causes speculation should be reduced. Instead of discussing the results achieved in the study, the possible positive or negative effects of the relevant results can be mentioned.

It is clear that there was a success in the study, but I suggest emphasizing which deficiencies in the literature will be filled with this success or for which application areas the material development process is better. Considering all these, I think that developing the Conclusion part will increase the impact of the article.

Taking all these points into consideration, my recommendation is that this article may be published in Applied Sciences after a minor revision.

Round 2

Reviewer 2 Report

The abstract should be revised again. The revised version has not any insignificant improvement. More data should be presented, thereby the abstract will be more interesting.

Author Response

Response to Reviewer 2 Comments

Point 1:

The abstract should be revised again. The revised version has not any insignificant improvement. More data should be presented, thereby the abstract will be more interesting.

Response 1:

This comment was very helpful in enhancing the quality of our paper. The abstract was revised again according to what the reviewer kindly pointed out.
